# An Update on Neurofibromatosis Type 1-Associated Gliomas

**DOI:** 10.3390/cancers12010114

**Published:** 2020-01-01

**Authors:** Mina Lobbous, Joshua D. Bernstock, Elizabeth Coffee, Gregory K. Friedman, Laura K. Metrock, Gustavo Chagoya, Galal Elsayed, Ichiro Nakano, James R. Hackney, Bruce R. Korf, Louis B. Nabors

**Affiliations:** 1Division of Neuro Oncology, Department of Neurology, University of Alabama at Birmingham, 510 20th Street South, Faculty Office Tower Suite 1020 Birmingham, Birmingham, AL 35294, USA; eacoffee@uabmc.edu (E.C.);; 2Department of Neurosurgery, Brigham and Women’s Hospital, Harvard Medical School, Boston, MA 02115, USA; jbernstock@partners.org; 3Division of Pediatric Hematology and Oncology, Department of Pediatrics, University of Alabama at Birmingham, Birmingham, AL 35294, USA; gfriedman@peds.uab.edu (G.K.F.); kmetrock@peds.uab.edu (L.K.M.); 4Department of Neurosurgery, University of Alabama at Birmingham, Birmingham, AL 35294, USA; gchagoya@uabmc.edu (G.C.); gelsayed@uabmc.edu (G.E.); inakano@uabmc.edu (I.N.); 5Department of Pathology, University of Alabama at Birmingham, Birmingham, AL 35294, USA; jrhackney@uabmc.edu; 6Department of Genetics, University of Alabama at Birmingham, Birmingham, AL 35294, USA; bkorf@uabmc.edu

**Keywords:** neurofibromatosis, gliomas, glioblastoma, therapeutics, astrocytoma

## Abstract

Neurofibromatosis type 1 (NF1) is an autosomal dominant tumor predisposition syndrome that affects children and adults. Individuals with NF1 are at high risk for central nervous system neoplasms including gliomas. The purpose of this review is to discuss the spectrum of intracranial gliomas arising in individuals with NF1 with a focus on recent preclinical and clinical data. In this review, possible mechanisms of gliomagenesis are discussed, including the contribution of different signaling pathways and tumor microenvironment. Furthermore, we discuss the recent notable advances in the developing therapeutic landscape for NF1-associated gliomas including clinical trials and collaborative efforts.

## 1. Introduction

Neurofibromatosis type 1 (NF1) is the most common tumor suppressor syndrome, with an incidence of approximately 1 in every 2500 to 3500 births [1]. NF1 is caused by pathogenic variants in the *NF1* gene, located at chromosome 17q11.2 [2,3]. Such variants can be familial or occur de novo, with the latter occurring in ~50% of individuals with NF1 [4]. Though NF1 is an autosomal dominant disorder with complete penetrance, there is vast variability in clinical presentation, even in monozygotic twins [5]. There are some genotype-phenotype correlations for specific NF1 variants [6,7], but much of the variability in phenotype has been attributed to stochastic events, environmental factors or modifier genes [8,9,10].

The *NF1* gene encodes neurofibromin, a cytoplasmatic 2818 amino acid protein that is widely expressed in the neurons and astrocytes of the central nervous system (CNS), as well as Schwann cells in the peripheral nervous system. Neurofibromin has been shown to control cell growth through two major intracellular pathways. First, neurofibromin negatively regulates the RAS pathway signaling through its action on GTPase-activating protein (GAP), stimulating the conversion of GTP-bound RAS to its GDP-bound form [11]. Increased RAS activity leads to the downstream activity of the MEK-ERK pathway as well as the PI3K-Akt-mTOR pathway (Figure 1) [12]. Neurofibromin has also been shown to positively regulate intracellular levels of cyclic adenosine monophosphate (cAMP), which in turn inhibits cell growth in some cells, including astrocytes [13]. Biallelic inactivation of *NF1* gene function is required for tumor formation; i.e., the somatic inactivation of the unaffected *NF1* allele is a “second hit,” which leads to absence of neurofibromin within affected cells [14,15].

## 2. Gliomagenesis in Neurofibromatosis Type 1

NF1 is associated with tumors of the peripheral and central nervous system (CNS). The most common CNS tumors in NF1 are gliomas, which are seen in approximately 20% of patients [16,17]. Gliomas usually affect children, with mean age at diagnosis of 4.5 years; the vast majority of such tumors originate within the optic nerves, optic chiasm, and/or hypothalamus. While individuals with NF1 are at higher risk for developing low-grade gliomas compared to high-grade gliomas [18,19], their risk for high-grade glioma is increased by 50-fold when compared to the general population [20,21]. Indeed, high grade gliomas are rare tumors and the reported higher risk in children and adults with NF1 is based on epidemiologic studies and several case series [22].

The World Health Organization (WHO) classification of gliomas has been refined and incorporated molecular parameters, namely 1p/19q codeletion, IDH1/2 mutation, and histone H3-K27M, in addition to histology to define many tumor entities [23]. In general, low-grade gliomas form a group of WHO grade I and grade II tumors while high-grade gliomas form a group of WHO grade III and IV based on malignancy grade, molecular markers and presumed cell of origin. The most common glioma associated with NF1 is pilocytic astrocytoma, a WHO grade I tumor, with the optic pathway glioma being a hallmark lesion [24]. Another low-grade astrocytoma that was reported in children with NF1 is pilomyxoid astrocytoma and the grading was suppressed in the revised 2016 WHO Classification to WHO grade I [25]. In contrast to pilocytic astrocytomas, diffuse astrocytomas, which form WHO grade II, III and IV tumors, are more common in adult individuals with NF with only 12% presenting before the age of 20 [26]. A clinicopathologic study that examined tumors from 100 individuals with NF1 reported pilocytic astrocytoma frequency to be 49% while diffuse astrocytoma to be 27% which included WHO grade II (5%), III (15%), and IV (7%) though this grading used the 2007 WHO Classification [26]. A recently published comprehensive genomic study performed in 23 high grade and 32 low-grade gliomas in individuals with NF1 demonstrated that children developed mostly low-grade gliomas (i.e., 77% of pediatric gliomas were low grade) whereas 78% of tumors in adults were high-grade [27]. The study included whole exome sequencing of tumor and matched blood germline DNA to identify germline and somatic single nucleotide variants, small insertions and deletions, and copy number variants. The *NF1* variants observed in germline DNA were typically truncating and led to frameshifts, which did not cluster into specific NF1 protein domains. There was no association between particular patterns of *NF1* genetic variants and the risk of developing glioma. The data supported prior reports that a “second-hit” is required to develop tumors [28], as loss of heterozygosity in the *NF1* region was detected in the majority of tumors. NF1-associated gliomas were found to have distinct genetic signatures, distinguishing them from those observed in sporadic gliomas, as well as noted to display different genetic landscapes when comparing low- vs. high-grade gliomas. For example, the isocitrate dehydrogenase (IDH) gene mutations (IDH1 and IDH2) are detected in more than 70% of sporadic low-grade gliomas and in the majority of glioblastomas arising from lower grade gliomas [29]. Indeed, individuals with gliomas harboring IDH mutations have better prognosis than those with IDH wild-type [30]. IDH mutations were not detected in gliomas associated with NF1 regardless of grade (Figure 2). This finding may, in part, explain the observation that astrocytomas behave more aggressively than anticipated in adults with NF1 [31]. Another example is that mutations in H3.3 histone genes, frequently found in sporadic pediatric gliomas [32], were absent in all samples regardless of age. Low-grade tumors exhibited fewer mutations that were over-represented in genes of the MAP kinase pathway, while high-grade tumors were characterized by a higher mutation burden and frequent mutations of *ATRX*, typically co-occurring with alterations of *TP53* and cyclin-dependent kinase Inhibitor 2A (*CDKN2*). DNA methylation assigned NF1-glioma to LGm6, a poorly defined IDH wild-type subgroup enriched with *ATRX* mutations, which may represent a point of therapeutic intervention, as previous studies have shown that loss of ATRX increases sensitivity to DNA-damaging agents [33,34]. Table 1 summarizes the common molecular differences between NF1-associated gliomas and the LGm6 subgroup of sporadic gliomas [27,35].

Approximately 50% of low-grade NF1-gliomas displayed an immune signature, T lymphocyte infiltration, and increased neo-antigen load, implying that such tumors may also be targeted via immunotherapies. Such results were confirmed via immunohistochemistry for the T lymphocyte markers CD3 and CD8: the T-cell infiltrates in high-immune NF1-gliomas included cells positive for granzyme B, the cytolytic effector that is upregulated on CD8+ T-cell activation, while B lymphocytes (CD20) and macrophages (CD68) were rare both in high- and low-immune groups [36].

Summary table listing the frequencies (%) of mutations seen in NF1-glioma as studied by D’Angelo, et al. (*Nat Med*, 2019), separated by high grade (Grade III–IV) and low grade (I–II). This molecular profile most closely correlated to the LGm6 subgroup of pan-glioma cohort from The Cancer Genome Atlas (TCGA) project (Ceccarelli, et al. *Cell* 2016), which is an IDH-WT group enriched with ATRX mutations.

TERT = TERT copy number variant gain in NF1-Glioma and TERT promotor expression in LGm6 group. ATRX = inactivation of ATRX from any mutation. CDNK2A = loss of copy number variant. TP53 = frameshift or missense mutation in both groups. PTEN = combination of missense and frameshift mutations in the NF-1 glioma group; missense and loss in LGm6 group. PIK3CA = missense and in-frame indel. NF1 = frameshift/truncating. BRAF = missense in NF1-glioma group, missense and frameshift in LGm6 group.

## 3. Optic Pathway Gliomas

Low-grade gliomas are the most common CNS tumors in the pediatric population, both in children with and without NF1 [37]. Multifocality and predilection for the optic pathways are features commonly associated with low-grade gliomas in NF1. Optic pathway gliomas (OPGs) are the most common brain tumors in individuals with NF1, with the majority classified as pilocytic astrocytoma (PA) [38].

### 3.1. Genetic and Molecular Pathophysiology

Due to the tumor location, OPG surgery is rarely performed; therefore, there is a relative dearth of genomic and/or microenvironmental studies given the lack of tissue. Multiple studies have been conducted to identify genotype-phenotype associations in NF1-associated OPGs, but reports are conflicting, probably due to the smaller sample sizes [39,40]. Studies have suggested that individuals with mutations in the 5′ tertile (exon 1-21) on *NF1* gene have a greater risk of developing OPG, but this was not confirmed in a subsequent study [41,42]. A large cohort study that examined *NF1* mutations in 215 NF1 patients (100 of them had OPGs) observed that those with variants in the cysteine/serine-rich domain of the *NF1* gene (CSRD, residues 543–909), which is located in 5′ tertile, had higher risk of developing OPGs [39]. A recent genotype-phenotype study reported a more severe phenotype in individuals with NF1 who carry missense mutations affecting one of five neighboring codons 844–848 located outside the GAP-related domain [43]. The study presented 162 individuals heterozygous for a constitutional NF1 missense mutation in one of the five neighboring codons 844–848. The cluster of the recurrent missense mutations reported in this study involving aa 844–848 is located within the CSRD domain, which is likely functionally important, and was originally described by Fahsold et al. [44]. The reported individuals have high prevalence of severe NF1 phenotype, including plexiform and/or spinal neurofibromas, symptomatic OPGs, skeletal abnormalities, and other malignant neoplasms.

Some studies of the tumor microenvironment have highlighted the role of microglia in OPGs, possibly due to the release of monocyte chemoattractant protein-1 (MCP-1) by the gliomas [45,46]. Microglia can have an immunosuppressive role in the tumor microenvironment, through the release of Tumor Growth Factor beta (TGFβ), and Vascular Endothelial Growth Factor (VEGF); findings that raise the possibility of immunomodulation of microglia as a possible therapeutic target in NF1-associated gliomas (Table 2) [47].

### 3.2. Clinical Presentation

Children with NF1, 6 years of age or younger, are at the greatest risk for developing OPG, with a slight female preponderance [48,49]. In contrast to non-NF-associated OPGs, NF1-associated OPGs are frequently asymptomatic and some may spontaneously regress [50]. Symptomatic OPGs are almost exclusively diagnosed in children younger than 8 years of age [51], and present with decreased visual acuity, visual field deficits, diminished color perception, optic nerve atrophy, and/or proptosis. Endocrinological problems, especially precocious puberty, may be seen with chiasmatic lesions [52]. It is noteworthy that young children rarely complain of visual loss; hence, reliable and reproducible measures to detect vision changes are necessary. One retrospective study of 54 patients with NF1-associated OPGs demonstrated that 59% had ophthalmological signs at time of presentation [53]. The signs included decreased visual acuity (72%), proptosis (31%) and, in one instance, nystagmus. Precocious puberty was reported in 12 (40%) children with chiasmal OPG, with accelerated linear growth being the first sign [54]. Though there are no current clear prognostic features for OPG progression, patient age, sex, and tumor location may predict disease course and influence treatment initiation. For example, post-chiasmatic tumors lead to vision loss in 62% of patients, compared with 32% in optic nerve and chiasmatic lesions [52]. Age at presentation can be of prognostic value; affected individuals who are younger than 2 years of age or 10 years of age or older at the initial presentation are more likely to have progressive disease that requires treatment [55,56].

### 3.3. Treatment

While approximately 20% of individuals with NF1 will develop OPGs, only 30–50% of these will be symptomatic and only one-third will require therapeutic intervention [57]. No clear correlation between the imaging features and the biological behavior of these tumors has been found [58]. Hence, close follow up of individuals with NF1-associated OPGs in NF centers using standardized visual assessment metrics is necessary to ensure that children with silent OPGs do not undergo treatment that can lead to unnecessary complications. Common widely available methods that provide objective visual field assessment include Snellen charts, HOTV charts, and Teller Acuity Cards. These methods along with optic disc pallor were evaluated as visual end points in clinical trials for NF1-associated OPGs [59]. Optic coherence tomography (OCT) is being tested to standardize the visual assessment in NF1-associated OPGs [60]. OCT provides an objective assessment of the retinal nerve fiber layer thickness and is a unique noninvasive tool to monitor children with OPG in whom traditional visual assessment is challenging [61]. Another objective noninvasive tool for visual assessment in NF1-associated OPGs is automated tractography of the optic radiations that was validated in a recent study [62]. Screening MRIs for NF1-associated OPGs are not indicated, as the decision to treat is based on clinical, rather than radiographic changes [63], though MRI provides a tool in children with NF1 in whom accurate assessment of visual acuity is not feasible.

In those with declining visual acuity, chemotherapy is considered the mainstay of treatment. First-line chemotherapeutic agents include vincristine and carboplatin [64], while second-line agents include vinblastine [65], vinorelbine [66] and temozolomide [67]. One report showed improvement in visual acuity after using bevacizumab in four cases of refractory OPG (two sporadic and two NF1-associated OPG) [68]. The aim for chemotherapy is to achieve stability and prevent further vision loss, as the currently used chemotherapeutic agents rarely restore premorbid visual acuity [69,70]. As in other tumor suppressor gene syndromes, radiotherapy is usually avoided in NF1-associated OPGs for concern of secondary tumors [71]. Highlighting the risk of radiation in children with NF1, a recent report demonstrated that among NF1-affected individuals with a primary tumor, the risk of secondary neoplasms was 2.8-fold higher in patients who received irradiation [72]. Another risk of radiotherapy in children with NF1 is the development of Moyamoya syndrome due to the radiation exposure to the circle of Willis blood vessels adjacent to the optic pathway [73]. Surgical excision of OPG is not feasible due to the tumor location and is usually reserved for instances of complete loss of vision, severe proptosis, or hydrocephalus [74]. In those with refractory OPGs, small molecule inhibitors have been used in clinical trials. Sorafenib, a multi-kinase inhibitor, is one of the tested agents, but the study was stopped due to unexpected accelerated tumor growth [75]. In this study, 11 patients were evaluated for response and only three had NF1. In vitro studies with sorafenib indicate that this effect is likely related to paradoxical ERK activation. A promising agent is selumetinib, which has shown favorable results in phase II studies of NF1-associated low grade gliomas [76].

## 4. Non-Optic Pathway Intracranial Gliomas

While OPG is commonly diagnosed in children with NF1, non-OPGs are frequently seen in slightly older children and young adults [26]. In a retrospective study of 104 individuals with NF1-associated CNS tumors, the brain stem was the second most common location of brain tumors after the optic pathway [31]. Brain stem tumors represented approximately half of all non-optic pathway gliomas in that cohort and were associated with less favorable prognosis. Although the cerebellum is the most common anatomical location for sporadic pilocytic astrocytomas, this site is rarely involved in NF1 [23]. While OPGs present in early childhood (mean age at presentation 4.5 years), brain stem gliomas tend to present in late childhood (mean age at presentation 7 years) [77,78]. The majority of gliomas presenting in adults with NF1 are high-grade tumors that arise most frequently in the cerebral hemispheres [27].

A histologically distinctive subtype of astrocytoma was recently described in individuals with NF1, which is similar to subependymal giant cell astrocytoma frequently observed in individuals with tuberous sclerosis complex (“SEGA-like astrocytomas”) [79]. In this study, Palsgrove and colleagues examined 14 tumors, 12 of them developed in individuals with clinical diagnosis of NF1. The average age at diagnosis was 28 years (range 4–60, 9 females, 5 males). All tumors occurred in the supratentorial compartment, with the frontal lobe being the most common site (*n* = 8). Tumors were predominantly low grade (*n* = 12). The tumors were moderately cellular and characterized by cells with plump eosinophilic/glassy cytoplasm and a large nuclei with mcaronucleolia resembling SEGA, no mutations identified in IDH1, IDH2, BRAF and H3K27M. All tumors expressed at least one glial marker (GFAP 10/10, S100 4/4, OLIG2 8/8) with variable positivity in neuronal markers, Moreover, the activation of the mTOR pathway was present to a greater extent than in other NF1-associated gliomas, a finding that was supported by prior reports and may provide novel therapeutic approaches (Table 2) [80,81].

There are limited data available on glioblastomas in individuals with NF1, with few cases reported in the literature [82,83,84,85]. Although it is challenging to draw a conclusion based on few case reports, the observed age of occurrence of NF1-associated glioblastoma in adults, mean of 38.3 years, is much younger than the mean age for patients with sporadic glioblastoma (Figure 2) [82]. Indeed, there is growing evidence that NF1-associated glioblastoma may comprise a unique subset of glioblastoma IDH wild type. Future large scale molecular and natural history studies are required to demystify the NF1-associated glioblastomas and how they relate to sporadic gliomas.

It is unclear if there is a significant difference in overall survival between individuals with NF1-associated and those with sporadic glioblastoma. A clinicopathologic study that identified five children with NF1-associated glioblastoma indicated better outcomes compared to sporadic glioblastoma [20], an observation similar to another study that identified four adults with NF1-associated glioblastoma [85]. Nevertheless, the overall survival remains poor in NF1-associated high-grade gliomas, with reports of enhanced toxicity when the standard therapy (gross total resection followed by fractionated radiotherapy, with concurrent and adjuvant temozolomide) is administered [86,87].

Combinatorial therapies against NF1-associated cancers, namely malignant peripheral nerve sheath tumors (MPNST), were first reported in 2011 by De Raedt et al. [88], and a phase II study using selumetinib (MEK inhibitor) and sirolimus (mTOR inhibitor) is currently active (NCT03433183). *NF1* loss is considered a marker that confers resistance to targeted therapy not only in NF1-associated cancers like MPSNT, but also in sporadic cancers like lung carcinoma, melanoma, and neuroblastoma [89,90,91]. In lung cancer models, reduced *NF1* expression mediates resistance to EGFR therapy, and blocking MEK restores the response [89]. A similar finding has been reported with resistance to BRAF inhibitors in *NF1*-mutant and *BRAF*-mutant melanomas, yet the tumors were sensitive to combined inhibition of MEK and mTOR [91]. These findings highlight the need for novel therapeutic approaches in NF1-associated high-grade gliomas through combinatorial therapies that target different intercellular pathways and the tumor microenvironment to overcome drug resistance imposed by NF1 loss [92].

## 5. Therapeutic Development

Loss of neurofibromin in glial cells leads to increased RAS activity and its downstream RAS effectors [93]. This has implications not only for NF1-associated tumors but in other cancers where mutation or overexpression of *RAS* genes occur, collectively known as RASopathies [94,95]. Also, somatic *NF1* mutation has been reported in 5–10% of sporadic tumors like glioblastoma, breast cancer, juvenile myelomonocytic leukemia, and lung adenocarcinoma [96,97]. The shared genetic and molecular landscape between individuals with NF1 and those with other sporadic cancers provides an opportunity for therapeutic development that may benefit NF1 and non-NF1 affected individuals alike [12]. Preclinical studies showing disease response to MEK inhibition have led to multiple clinical trials in NF1-related lesions [98,99]. Targeting RAS-downstream pathway signaling may provide opportunities for synergy across NF1 manifestations, as shown preclinically with MEK inhibition in plexiform neurofibromas, gliomas, and bone pathology [100,101,102]. To date, at least four MEK inhibitors have progressed to clinical trials in NF1; mirdametinib (formerly PD-0325901), trametinib (GSK1120212 Mekinist), binimetinib (ARRY-438162, MEK 162), and selumetinib (AZD6244). In February 2018, selumetinib, co-developed by AstraZeneca and Merck & Co., received breakthrough status from the FDA. Selumetinib was granted Orphan Drug Designation based on data from the phase II trial that tested selumetinib in pediatric patients with inoperable plexiform neurofibromas (NCT01362803) [103]. Selumetinib is currently in phase II study for low-grade glioma and phase III study for OPG.

While MAPK/ERK activation has been seen across all astrocytoma types, including sporadic and NF1-associated astrocytomas, MAPK/ERK activation in NF1-associated astrocytomas is independent of *BRAF* alteration [104]. The majority of non NF1-associated pilocytic astrocytoma carries an activation of *BRAF* due to fusion of *BRAF* with *KIAA1549* [105]. Paradoxical MAPK and mTOR activation was reported in *BRAF* fusion cell lines with first generation BRAF inhibitors [106]. A phase II trial with sorafenib, an inhibitor targeting BRAF, VEGFR, PDGFR and c-kit was terminated due to unexpected acceleration of tumor growth, which was not dependent on *NF1* status [75].

Immunotherapy, including treatments such as checkpoint inhibitors, vaccines, and oncolytic virotherapies, has emerged as a promising therapeutic approach for a variety of cancers [107,108]. The adaptive resistance to cell-mediated immunity may play a major role in NF1-associated tumors microenvironment through the expression of programmed death ligand 1 (PD-L1) on tumor cells and the presence of tumor infiltrating lymphocytes [109,110,111]. These emerging reports suggest that NF1-associated tumors, including gliomas, may be responsive to immunotherapy, which should be explored in clinical trials.

Due to the major differences between NF1 and non NF1 associated tumors in relation to natural history, disease manifestations and prognosis, there is an unmet need to establish standardized endpoints in NF1 clinical trials. These standardized endpoints would allow precise data interpretation and the assessment of efficacy across different studies. That has led to the establishment of The Response Evaluation in Neurofibromatosis and Schwannomatosis (REiNS) International Collaboration in 2011 to achieve consensus among NF community regarding the clinical trials endpoints and designs [112], which highlights the fundamental role of the collaborative efforts among academic, federal regulatory and patient advocacy organizations in drug development in NF1. These organizations include the National Institute of Health, the Department of Defense Congressionally Directed Medical Research Program (CDMRP), the Children’s Tumor Foundation and the Neurofibromatosis Therapeutic Acceleration Program at Johns Hopkins University [113].

## 6. Conclusions

In summary, the NF1 tumor-predisposition syndrome is associated with a heterogenous pattern of gliomas with distinct genetic signatures which differ from sporadic gliomas. Undoubtedly, the past two decades have witnessed a paradigm shift in NF1 therapeutic development, thanks to an increased understanding of the pathogenesis and molecular landscape of the NF1-associated tumors. As we enter the era of precision oncology, clinical trials of new drugs targeting the RAS/MAPK pathway are underway for NF1-associated gliomas, and other immunotherapies are under development. The academic, federal regulatory, and patient advocacy communities are collaborating to develop consensus endpoints for NF1 trials and to establish a clear pathway for developing measurement tools to support drug approvals for NF-associated tumors, which ultimately will contribute to improved outcomes for NF1-associated gliomas.

## Figures and Tables

**Figure 1 cancers-12-00114-f001:**
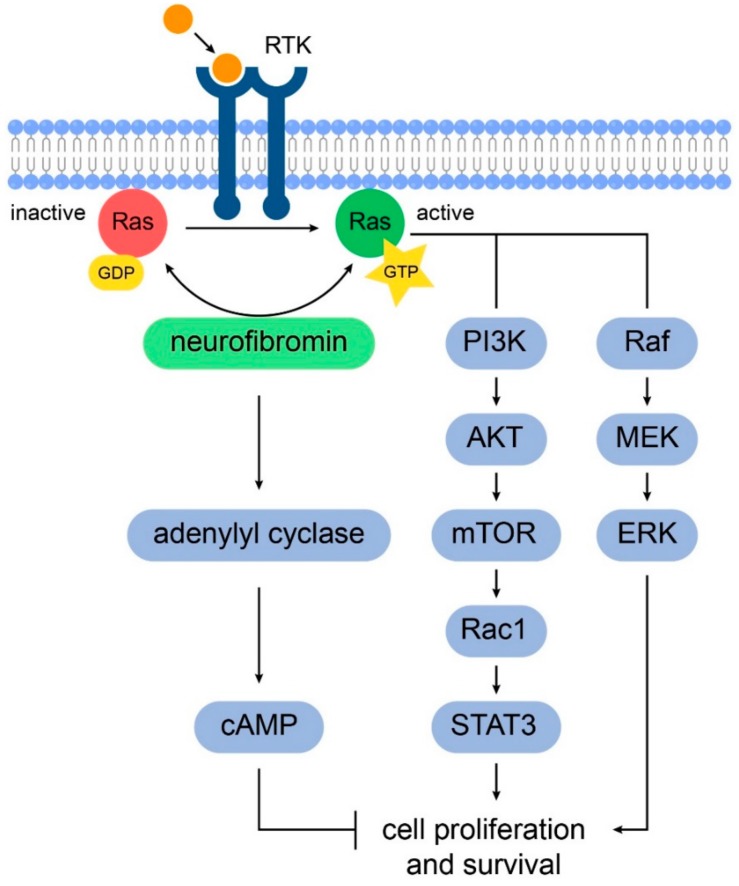
Schematic representation of the signaling pathways involved in NF1 tumorigenesis. Neurofibromin positively regulates adenylyl cyclase to increase intracellular cAMP levels which inhibits glial cell proliferation and survival. Also, neurofibromin promotes the conversion of active GTP-bound RAS to its inactive GDP-bound conformation. In NF1, the increased RAS activity in astrocytes leads to cell proliferation through the downstream activation of the PI3K/AKT/mTOR and RAF-MAK/MEK pathways.

**Figure 2 cancers-12-00114-f002:**
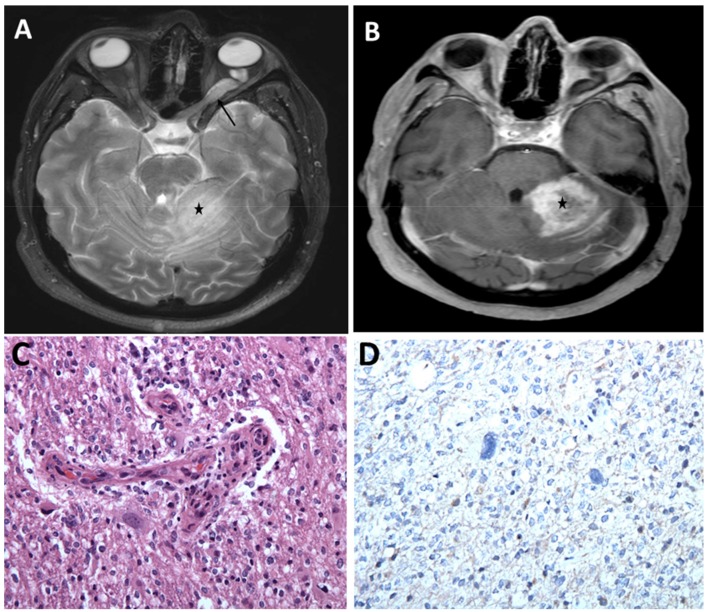
Optic pathway glioma and a high-grade cerebellar glioma in a young adult with NF1. (**A**) MRI brain, axial T2 sequence showing hyperintense left optic nerve lesion (arrow) and ill-defined hyperintense lesion within the left cerebellum (asterisk) associated with mass effect. (**B**) Post-contrast T1 sequence showing heterogeneous enhancement of the left cerebellar lesion concerning high grade neoplasm (asterisk). Histopathologic evaluation of the left cerebellar lesion was consistent with glioblastoma, WHO grade IV, IDH wild-type (**C**) Infiltrating glioma exhibiting atypical cells and vascular endothelial proliferation (H&E, 200×). (**D**) Tumor cells are negative for IDH1 (R132H) mutant protein (IHC, 200×).

**Table 1 cancers-12-00114-t001:** Somatic and Germline alterations in NF1-Glioma compared to the LGm6 subgroup of sporadic gliomas.

Variation	*NF1-Glioma*		*LGm6 Sporadic Glioma*		
	High Grade	Low Grade	High Grade		Low Grade
			Grade IV	Grade III	Grade II
*IDH Wild-Type*	100	100	100	100	100
*TERT*	47	12	43		
*ATRX*	38	3	13	42	0
*CDNK2A*	58	19	59	46	17
*TP53*	29	0	35	42	0
*PTEN*	12	0	54	38	0
*PIK3CA*	17	0	13	0	8
*NF1*	88	91	22	50	8
*BRAF*	0	3	4	0	15
*NF1 germline mutation*	92	91			

**Table 2 cancers-12-00114-t002:** Clinical trials for NF1 associated gliomas.

Drug	Target	Tumor	Phase	Age	Endpoints	Status
Vinblastine +/− Bevacizumab NCT02840409	Cytotoxic/VEGF	LGG	II	6 months to 18 years	Response rate, OS, PFS, visual outcome measures, OCT	Recruiting
Pegylated interferon NCT02343224	Tumor microenvironment	PA or OPG	II	3 to 18 years	Response rate	Recruiting
Pomalidomide NCT02415153	Angiogenesis/immunomodulation	NF1-associated CNS tumors	I	3 to 20 years	Toxicity, MTD	Active, not recruiting
Lenalidomide NCT01553149	Angiogenesis/immunomodulation	PA or OPG	II	0 to 21 years	Response rate	Active, not recruiting
Everolimus (RAD0001) NCT01158651	mTOR	LGG	II	1 to 21 years	Response rate	Active, not recruiting
Binimetinib (MEK162) NCT02285439	MEK	LGG	I/II	1 to 18 years	MTD, response rate	Recruiting
Binimetinib (MEK162) NCT01885195	MEK	Solid tumors with *NF1* mutation	II	Older than 18 years	Response rate	Completed (pending results)
Selumetinib NCT01089101	MEK	LGG	I/II	3 to 21 years	Safety, MTD, Response rate	Recruiting
Selumetinib (Selumetinib vs. carboplatin and vincristine) Randomized NCT03871257	MEK	OPG	III	2 to 21 years	Event-free survival ∗, visual acuity	Not yet recruiting
TAK-580 NCT03429803	RAF (pan-RAF kinase inhibitor)	LGG	I/II	1 to 18 years	Toxicity, MTD, 6-month PFS	Recruiting

Abbreviations: LGG, Low-Grade Glioma; MEK, mitogen activate protein kinase; MTD, maximal tolerated dose; mTOR, mammalian target of rapamycin; OPG, Optic-Pathway Glioma; OS, overall survival; PA, Pilocytic Astrocytoma; PFS, progression free survival; RAF, Rapidly accelerated fibrosarcoma; VEGF, vascular endothelial growth factor. ∗ Event-free survival is the time frame from randomization to the first occurrence of any of the following events: clinical or radiographic disease progression, disease recurrence, second malignant neoplasm, or death from any cause, assessed up to 10 years.

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
