# Peer review of "An Update on Neurofibromatosis Type 1-Associated Gliomas"

_cancers, 2020, doi:10.3390/cancers12010114_

Round 1

Reviewer 1 Report

In this study, Lobbous and coll. review NF-1 associated gliomas.

This review is interesting and can provide an overlook of these tumors to the readership. However, I suggest some changes.

In paragraph 2, the authors state that NF-1 gliomas do not show IDH1 mutation. However, they should also mention IDH2 mutation, which can be present in gliomas. In addition, IDH mutation in low-grade gliomas is associated with the histotype (it is absent in pylocitic astrocytoma).  I would add a paragraph (and Table), summarizing which histotypes of gliomas (according to WHO Classification) can be associated with NF1 (e.g. pylocitic astrocytoma? diffuse astrocytoma? glioblastoma?) The paragraph on SEGA-like astrocytoma could be expanded, with mention of histopathological features, localization, etc. Table 1. A column with tumor type should be added

Reviewer 2 Report

The authors provided a review of literature concerning the pre-clinical and clinical spectrum of intracranial gliomas in patients affected by neurofibromatosis type 1.

The mechanisms of gliomagenesis as well as the role of molecular pathways and tumor microenvironment have been discussed. Moreover, recent advances concerning possible pharmacological treatment of NF1-associated gliomas have been considered.

The paper is well written and most of the topics covered have been well organized and adequately described.

The only criticism concerns the description of genetic features of patients with NF1-associated- and sporadic- gliomas.

A table summarizing data including the mutated genes at both germline and somatic level in patients with sporadic and NF1-associated gliomas can provide a comprehensive overview of the current knowledge about the genetic features of gliomas in the two groups of patients, helping the reader to carry out informative comparisons to better understand the pathogenic mechanisms underlying gliomas.

Furthermore, the reported clustering of NF1 mutations in CSRD domain should be discussed attempting a pathogenic significance identification.

Round 2

Reviewer 1 Report

The authors addressed all comments arisen.